# Towards an open and decentralized case law curation ecosystem

**Eleni Panagou, Manolis Vavalis** ¤*

Department of Electrical and Computer Engineering, University of Thessaly, Volos, Greece

¤ Current address: Mathematics Department, University of California, San Diego, California, United States of America

* mav@uth.gr

## Abstract

Case law is the term that refers to reports of past court decisions. It is considered an essential source of law, vital for legal professionals. Existing case law services are currently centralized, with an entity having complete control over the data and often charging fees for its access and other adding value services. This paper attempts to leverage the potential of blockchain technology in order to develop a public and decentralized platform that allows the submission of court decisions in a decentralized database and employs a network of curators who offer their validation, classification, and evaluation. Specifically, we design, analyze and implement *AnyCase*, a proof-of-concept prototype system on the Ethereum platform. We focus on the establishment of a sybil-resistant voting protocol used for reaching agreement and the development of a tokenized economy that incentivizes participation. Our preliminary analysis indicates that, besides being decentralized, *AnyCase* has the potential to compete with existing centralized systems in several other aspects.

**Data Availability Statement:** All relevant data are within the manuscript.

**Funding:** The authors received no specific funding for this work.

## Introduction

The essence of case law is captured by the principle of *stare decisis*, which is Latin for "to stand by decided matters". This principle compels courts to adhere to past rulings in order to avoid contradicting themselves when applying the law. In other words, cases that present similar facts or similar key points should be treated the same. Consistency is of utmost importance. Courts are not only discouraged from contradicting other courts' decisions, but are also outright forbidden from doing so when faced with the decision of a higher or equally ranked court in the hierarchy of the jurisdiction in question. The reason for adhering to this principle is the aspiration of maintaining a standard enforcement of the law.

It is common for the interpretation of the law to be unclear, as statutory language tends to be ambiguous [1] and at times the legality of a given act can be uncertain. Also, when considering extraordinary cases, it is reasonable to assume that legislation could be insufficiently developed and thus incapable of offering a straightforward resolution to the matter. In these cases, case law, if available, becomes the catalyst for the case at bar as well as future cases of the same kind. Since courts are obligated to avoid contradictions with past rulings, case law constitutes

**Competing interests:** The authors have declared that no competing interests exist.

an indispensable tool for legal professionals and is therefore both an extension and a source of law itself.

The provision of reliable legal information for attorneys and law firms in a consistent, digitized format has been the focus of several companies in recent years. Services such as LexisNexis Case Law [2], WestLaw [3] and Bloomberg provide comprehensive collections of case law. They employ attorney editors who review, analyze and summarize cases. Others, such as Luminance [4] and Eigen Technologies [5], a startup funded by Goldman Sachs, utilize AI in order to automate the extraction of useful data from legal documents. Finally, there are also services which strive for free and open access to case law for the public, such as Google Scholar, and the Caselaw Access Project [6], which exposes an API endpoint for this purpose.

Multiple issues arise with the approaches above, the most crucial of which is their high degree of centralization. A central authority has complete control over the data as well as the ability to restrict access to it. Restrictions may be applied by allowing minimal or no access to data without a subscription, while subscription fees may be freely raised and access to the data may be revoked at any time. Additionally, a centralized service presents a single point of failure that could fall victim to attacks and censorship.

Furthermore, [7] mentions that automatic categorization of case law documents can be challenging due to their complexity. As a result, the larger part of legal research services employ attorneys for the curation and quality control of the massive collection of documents they provide, a procedure which undoubtedly requires expert knowledge and immense effort. This leads to duplication of effort and a scatter of legal information across different sources and inconsistent formats, since no standard metadata structure has been agreed to represent a case law document. Lastly, as the volume of digitally available legal documents increases, the need for a more scalable and robust solution is apparent.

To this day, there is no decentralized, unified, comprehensive, aggregator of case law which is open access and offers opportunities for contribution by anyone. The open, trustless, immutable and publicly verifiable nature of a public blockchain could render it an essential component for the establishment of such a platform.

This paper aims to design and implement a public and decentralized case law ecosystem where the provision and curation of legal knowledge is a collaborative effort of an entire network of contributors. The ecosystem will attempt to enhance the transparency and ease of access to verified and categorized legal information, as well as eliminate single points of failure by relying on decentralized principles and components.

The regulations that define the ecosystem are permanently stored in a ledger as a set of smart contracts and enforced through the trustless execution offered by the Ethereum blockchain, which constitutes the cornerstone of the platform. The submission of court decisions is open to any user, while the validation and labeling of newly submitted decisions according to their legal subject and juristic value are performed through timed polls, which are open to any and all users of the platform. The influence of each user is measured by parameters such as their financial investment and positive contributions. Appropriate behavior is encouraged through financial incentives, which are formed through reward distribution using a digital token. Autonomy is maintained by employing the collective wisdom for reaching agreement, ensuring that no intervention by a central authority is required for the successful operation of the platform.

Consequently, the adoption of this platform on a national basis could potentially be of great value not only to legal professionals but also to any parties interested in case law research regardless of their legal expertise. Additionally, the establishment a tokenized economy will directly benefit the platform's contributors. At the time of writing, a thorough search for decentralized applications of such type did not yield any results.

The rest of the paper is organized as follows. Next, we provide an overview of the background and fundamental technologies addressed throughout the paper. Section Related work presents blockchain efforts and the associated technologies that may provide support for the development of our system. In section Design, an overall case law system design is presented and analyzed. Section Implementation discusses implementation issues and section Synopsis and prospects concludes the paper and offers recommendations for potential enhancement of the platform's capabilities.

## Background

Next, we offer a brief description of the Blockchain technology and we discuss important elements of the Ethereum platform as it is needed for understanding our design, analysis, and implementation of our system.

Blockchain is a peer-to-peer (P2P) network that maintains an immutable record of digital transactions. Transactions are signed messages that are propagated within the network and affect the global state of the chain. They are validated and grouped into cryptographically secured *blocks*. The *genesis* block is the first block of the chain and each subsequent block is appended through the hash of its previous block. This creates a permanent and tamper-resistant data structure.

In a public (or *permissionless*) blockchain, anyone may participate in the block creation process by connecting to the network with the chain's client software. This software implements the blockchain's specification and allows a peer to synchronize with the network by downloading a complete copy of the blockchain, from the genesis to the most recently generated block and begin participating in the validation (consensus) process.

Out of the various existing block validation schemes, the most widely used is the *Proof-of-Work* (PoW) [8]. In a PoW blockchain, the consensus is reached by competition. Peers participate in a process called *mining*, during which they use their computational power to solve a puzzle, which is usually a resource-intensive hashing process. The puzzle must be difficult to solve but easy to verify, and rapidly determine the miner's claim. The first miner to provide the proof of its solution is authorized to append the next block to the chain and in return, it gains an amount of *cryptocurrency*, the digital currency managed by the network. Miners compete to solve the puzzle faster than other miners in order to reap the rewards. Consequently, the blockchain is economically secured with cryptocurrency, since miners act beneficially to the network out of self-interest.

Proof-of-Stake (PoS) is an alternative consensus algorithm that is based on peers staking amounts of cryptocurrency in order to become validators and as such earn the ability to propose new blocks or vote on whether a proposed block is valid. Voting power is proportional to the staked amount, meaning the amount of cryptocurrency locked into a deposit by each validator. The validators are chosen with a round-robin protocol. If their proposed block is accepted by the majority of validators, they gain a reward, based on their stake, while they can lose their stake if their proposed block is rejected.

### Ethereum and other blockchain platforms

There exist a plethora of currently operational blockchain platforms and the reader is referred to [9] for a related broad and up to day overview. Out of all these platforms we have carefully considered the ones which enjoy characteristics that required in our design and implementation. Namely Bitcoin, Hyperledger Fabric [10], EOS [11] and Ethereum.

Bitcoin regardless of its remarkable success and wide acceptance, it does not provide the required support for smart contracts. EOS uses a Delegated Proof-of-Stake (DPoS) model, in

which solely twenty-one validators are responsible for producing blocks. This does not constitute a purely decentralized solution. Hyperledger Fabric, supports the effective development of a blockchain tailored for a particular purpose and thus offers the highest amount of flexibility. It is mostly aimed towards private or *permissioned* blockchains. Ethereum is currently the dominant blockchain platform in particular if smart contract mechanisms concerns. It is considered as the de facto standard for building decentralized applications. Due to its relative maturity compared to other solutions, as well as its overall wider use and its huge development community, it offers superior infrastructure for aiding the development of applications like the one considered in this study. The aforementioned points solidified our decision to opt for Ethereum as our tool/platform of choice.

Ethereum [12] is a Proof-of-Work blockchain that allows the execution of programs called *smart contracts* in an auditable, transparent and trustless environment. The blockchain is used as storage for the global state of the system, while a consensus algorithm allows state synchronization within the peers of the network. Smart contracts are written in high-level programming languages and compiled to bytecode. They can then be executed on the *Ethereum Virtual Machine* (EVM), a fully sandboxed and isolated runtime environment. It has no access to other processes, the network stack or the file system as it is entirely virtual. Therefore code execution is completely deterministic.

Every Ethereum node has an instance of the EVM which is used to execute smart contract code triggered by transactions. The EVM has its own instruction set and is able to access block and account information such as addresses and their respective balances, number and timestamp of the latest block.

The following two account types are available in Ethereum.

**Externally owned accounts** which are owned and controlled by individuals through a public and private key pair. They can use their private key to sign and send transactions to other externally owned accounts or contract accounts, and they can receive Ether through their public key (or *address*). The generation of a new identity occurs locally on a user's machine and has virtually no cost. Thus, users may have one or more accounts at their disposal.

**Contract accounts** represent the address through which a smart contract may be invoked. Contract accounts can not send transactions independently. Their execution is triggered by a transaction originating from an externally owned account or another contract account.

Ethereum transactions are cryptographically signed messages from an externally owned account to another account. They can be used to transfer cryptocurrency named *Ether* to another account, invoke contract code or deploy a smart contract to the network. Every account has an Ether balance.

Ethers, are acquired as a reward for successfully mining a block and are used by the network as a means of paying for computation. Ethereum's smallest currency denomination is called *wei* and equals to $10^{-18}$ Ether.

**Gas.** Solidity, the most widely used contract programming language for Ethereum, is Turing-complete. This implies that contract code written in Solidity can be non-terminating, which, in the Ethereum world, can essentially perform a Denial of Service (DoS) attack on the network. This is because all peers that attempt to validate the program will inevitably be running it forever.

For this reason, Ethereum employs *gas*, a mechanism for controlling the computational resources a transaction attempts to consume. Every EVM instruction has a predetermined cost measured in gas units. Thus, gas represents the amount of work a miner needs to do in order to validate a transaction and include it in a block. Gas is purchased with Ether for every

transaction, and it is consumed as code execution progresses. Any unused gas is returned to the transaction sender. The sum of all transaction gas fees is sent to the miner of the block as a reward. If the gas sent with the transaction does not cover the cost of the operations needed to complete its execution, the transaction fails with an "Out of Gas" exception, any changes to the state are reverted and the price paid for the gas is lost, since the network already used computational power for the execution. Most wallets, which are responsible for constructing the transaction object, calculate the amount of gas needed to complete the transaction and set the correct amount for the user in the transaction headers.

The cost of gas is not fixed. The transaction sender is able to set the *gas price* of the transaction, which represents the conversion rate of ether to gas. It is thus used to calculate the amount of Ether to be paid for the gas spent during transaction execution. Miners naturally prefer to include transactions with higher gas price in their blocks due to gaining higher rewards for doing so. Thus, setting a higher gas price will result in a faster transaction confirmation time.

**Smart contracts.** Ethereum provides the ability to instantiate and execute programs in a trustless manner. Smart contracts are code that has been stored on the blockchain and can be invoked by anyone who has the ability to send transactions to the network. When a transaction is sent, it is stored in a transaction pool and all peers in the network individually execute the contract code in order to agree on its output.

Smart contracts are written in high-level languages such as Solidity and Vyper, compiled to EVM bytecode and deployed to the network. Once deployed, they acquire their own address through which their functions and storage are accessible. They are permanent and immutable: the only way to update a smart contract is to redeploy it. Furthermore, contract code cannot run autonomously. An account has to sign and send a transaction which includes all the input needed to invoke contract code in order to notify the nodes of the Ethereum network to execute it. As mentioned in Gas this does not come without cost.

All input and output of a contract is public knowledge and can be accessed by the whole network. Any sensitive input will be revealed as soon as the transaction is sent. A *commitment scheme* allows one to temporarily hide a chosen value stored in a public environment and having the ability to reveal it at some point in the future. Commitment schemes exploit the fact that hashing is a one-way operation and are performed in two phases. During the *commit* phase, a user publishes the hash of the desired value, while during the *reveal* phase, they publish the value itself. It is then possible for anyone to verify if the hash of the revealed value is equal to the committed hash. In that case, the user has successfully hidden their value during the commit phase and has proven that they have revealed the correct value. Applications of commitment schemes include secure coin flipping, zero-knowledge proofs, and secure computation.

**Tokens.** The permanent, immutable and trustless nature of the blockchain makes it a suitable option for asset management. Programmable assets can be implemented on top of Ethereum through smart contracts for cases when Ethereum's native cryptocurrency is not sufficient. An asset may be abstracted and implemented on the blockchain as a *token*. Tokens may represent currency, physical or digital resources, voting power, or access rights. There is absolute freedom to establish custom rules of ownership as well as asset management such as transfers, purchases and sales, while maintaining trustlessness and transparency.

Various token standards have been introduced that cover basic token concepts. They define an interface, meaning a set of methods which have to be implemented to cover the specification of the token. ERC20 [13] is the most widely used token standard. It represents ownership of value which can be divisible and interchangeable, much like real-world currency. The ERC721 [14] token standard is used to represent *non-fungible* tokens that denote ownership of

a unique, non-divisible asset or collectible, or even track negative occurrences such as loans. Every token is represented by a 256-bit identifier and its ownership can be transferred from address to address.

Token specifications determine only the minimum functionality that needs to be implemented. They can be easily extended by implementing more functions in order to cover more specific needs. Compliance with standards allows tokens to automatically be supported by various exchanges, other smart contracts, and applications.

Tokens may be created out of thin air in a process called *minting* and can then be exchanged for Ether through crowdsale contracts. Crowdsales can be programmed to have a certain duration, maximum supply amount or even form bonding curves so that the rates scale with the current supply, encouraging the contribution of early investors.

**Oracles.**   Smart contracts have no access to external data since the EVM in which they are executed is completely isolated from the real world. However, they may often require data from external systems. Use cases include true random number generation, identification, and authorization through decentralized identifiers (DIDs), access to online information such as market data, IoT sensor data such as temperature, or any events and triggers that happen off-chain but need to be acknowledged on-chain.

This data can be introduced to a smart contract via an *oracle*, a third-party agent which seeks, verifies and provides external information and events which originate from outside of the blockchain and are used in smart contracts. Usually, a smart contract is created by an oracle that obtains data from an external source and stores the information in its contract storage. Other smart contracts may access this information through the oracle contract. In addition, oracles can be part of multi-signature contracts that require, among others, the signature of an oracle in order to trigger a specific action, such as the release of funds.

Off-chain occurrences cannot be verified by the Ethereum network, thus depending on knowledge originating from, oracles constitute an inevitable security risk. For instance, let us assume the existence of an oracle that generates random numbers for a contract that implements a game of roulette. There is a high incentive to compromise or bribe the oracle if the financial reward for doing so will cover the cost of the attack. For this reason, oracles are, ideally, systems that rely on decentralized principles for increased security and transparency.

## Decentralized applications

Armed with publicly verifiable, trustless and guaranteed correct program execution provided by the Ethereum blockchain, smart contracts can be used as building blocks to develop decentralized applications. These applications, also known as dApps, use smart contracts to apply their business logic and implement a web interface that interacts with the blockchain using the `web3` stack. Optionally, they may also use a decentralized storage service, a decentralized messaging protocol or both. Web3 or "The Third Age of the Internet" [15] is the term used to describe the shift of focus of web applications towards more decentralized protocols, which exploit the autonomy and censorship-resistance provided by blockchain technologies.

The degree of decentralization of an application can vary. Some applications choose to utilize the blockchain only for a small part of their business logic, while others attempt to achieve complete decentralization by not only storing their data in decentralized file systems but also serving their web interface through them.

**Decentralized file services.**   Since the blockchain is an append-only data structure, storing data on it will eventually cause it to grow insurmountably, increasing synchronization time as well as block propagation time, thus reducing the overall performance of the network. As a result, there is an incentive to limit the amount of data stored inside contracts. This is achieved

by applying prohibitively large gas fees for storing data. For reference, the ADD opcode, which performs an addition operation, consumes only 3 gas units, while the SSTORE opcode, which saves a word (256 bits) to the contract storage consumes 20000 gas units [16].

Consequently, applications that use large amounts of data are forced to use contract storage in conjunction with an off-chain database to fulfill their storage needs. Contracts cannot access this storage, thus the application needs to sensibly split its data into contract and external storage.

Using any standard, centralized database to store external data is certainly possible. However, this approach suffers from the same issue that the application is trying to solve by using the blockchain: centralization. Public and decentralized file hosting services are able to meet most of the storage needs of a dApp without compromising decentralization and transparency.

The InterPlanetary File System (IPFS) [17] is a public, peer-to-peer global file hosting service. In the IPFS protocol, an uploaded file is split into small chunks of 256Kb which are then hashed and stored on multiple different nodes on the network. The hashes are stored in a distributed hash table (DHT), which is used to retrieve a file whenever it is requested. Storage provided by IPFS is content-addressable, meaning that a file is identified and accessed solely by its hash.

Additionally, the more popular a file is, the faster it can be accessed since it is replicated on multiple nodes. Conversely, a file that is not regularly accessed is in danger of being garbage collected from the network. At least one node has to be online and *pin* the file in order to guarantee availability. Pinning ensures that the file is always kept locally on the node and is never garbage collected.

Due to the above, IPFS is currently not able to ensure data availability. However, there has been work in creating incentives for monetizing permanent storage. Ethereum Swarm [18], which functions similarly to IPFS, is part of the web3 stack and works as an incentive layer for sharing permanent storage by rewarding Swarm nodes who offer their storage for use by other clients. Among others, Storj chain [19], Sia (https://sia.tech) and Filecoin [20] offer the same service but are built on top of their own dedicated blockchain and use their own native currency for reward distribution.

A much-desired property of content-addressability is the fact that immutability is guaranteed. Any changes to the file would result in it having a different hash. A commonly used pattern in the development of decentralized applications is to store large data on IPFS, obtain its unique content-addressed hash and permanently store it on the chain.

## Related work

As already mentioned in the Introduction, there currently exist several case law services. Their characteristics in general vary but they all fully rely on one or more central authorities who act both as fundamental cornerstones of the system and as primal operators and administrators. To the best of our knowledge, there exist no decentralized and public owned system for the needs of Case Law yet.

### Blockchain voting

Next, we review numerous protocols and applications that attempt the decentralization of the voting procedure using the blockchain. Emphasis is given in ensuring privacy of the vote during vote casting in order to eliminate bias.

In the voting protocol proposed in [21] the creator of the poll has the capability to grant a user the right to vote by registering her wallet address. The voters have a specific deadline

before which they are obligated to declare their vote. Only one vote per user is allowed for each poll.

The voting application given in [22] is designed for high-level educational instructions. Every user that can be identified with a valid identification code and email address that belong to the institution, has the ability to create, or participate in polls. This application combines Ethereum smart contracts and Paillier Homomorphic Encryption in order to ensure the privacy of the vote.

An attempt to enforce a single vote per user per poll is presented in [23]. It suggests the incorporation of an Ethereum light client in a mobile application and the registration of each voter in the system by utilizing the MSISDN (phone number) of the user's SIM card.

It is noted that none of the above solutions are completely decentralized since they rely on a central authority which the potential voters have to register to. Consequently, the maximum number of voters is known before the start of the poll and voters potentially compromise their anonymity [22, 23] since they are uniquely identified through personal information. Please note that, [21, 23] do not attempt to achieve privacy of the vote. Since submitted votes are visible to anyone on the chain, in order to guarantee confidentiality and non-repudiation, the votes need to be submitted to the contract in a hidden but verifiable format [22] or by using a commitment scheme.

**Commitment schemes.** Commitment schemes are valuable for temporarily hiding data in trustless environments. They are used in a plethora of applications on the blockchain, such as voting, blind auctions and creating randomly generated numbers for multiplayer games.

For example, the implementation of a particular commitment scheme [24], utilizes an oracle for random number generation by using the input of multiple participants. Each participant must send the hash of a secret number along with a pledge. After the submission phase is over, users reveal their secret number. In the final phase, the contract computes the random number by combining all valid inputs and returns the pledge to each participant that revealed a number whose hash matches their previously committed hash.

In the secure and self-tallying voting procedure described in [25] eventually the votes are publicly accessible and anyone may compute the tally of the election. A zero knowledge proof (a method by which one party can prove to another party that they know a value x, without conveying any information apart from the fact that they know the value x) is used to protect data integrity and authenticate users before determining the result of the election. A voter registers for an election by submitting a deposit and a zero knowledge proof for authentication purposes. The voting procedure is then performed using a standard commitment scheme for privacy preservation. An administrator orchestrates the poll by triggering each next phase after all members have participated in the current one. Revealing a valid vote returns the deposit to the user. If one or more voters do not commit their vote before the commit phase ends they receive their deposit back. After the last vote has been revealed, the administrator calls a contract method which tallies the votes.

The main drawback of this method is that an administrator entity is required to advance through the poll phases, which sacrifices autonomy. Furthermore, it requires knowledge of the number and identities of voters beforehand and it relies on all users cooperating in order to advance to the next phase.

For our needs, more flexibility is required regarding the number of voters in each poll. New voters may join the platform at any time, and voters may abstain from voting on some polls. It is important to recall that, regarding the Ethereum network, there is virtually no cost for generating new accounts and attempting to impersonate multiple voters. In these circumstances, when voter registration and identification is not an option, the result of the poll should not be computed by tallying individual votes, but by tallying voting power.

**Token-weighted voting.** The use of tokens as a voting right is justified by the assumption that the benefit of the platform is of direct interest to its stakeholders which have made financial investments in it by purchasing and holding tokens. Consequently, the larger their investment is, the heavier their vote should be. Variations of token-weighted voting are used by decentralized autonomous organizations (DAOs) to implement self-governance.

An implementation of token-weighted voting is presented in [26]. The votes are concealed through a commitment scheme in order to ensure confidentiality. Users can decide how many tokens they wish to lock in a specific poll. They may lock their tokens in multiple polls simultaneously. The amount of tokens they lock in a poll is used to calculate their weight for the poll in question. They may only retrieve their locked tokens after the end of the voting process, in order to prevent double-voting using the same tokens in different polls.

A blockchain governance method which allows users to vote with a token locking mechanism is described in [27]. In order for a proposal to be approved, first it needs to receive enough approval votes within a certain time window. If this is achieved, a new voting round starts where opponents of the proposal have to counter the approval using the same voting method. This *ping-pong* procedure continues until one side is not able to gather the needed votes to satisfy the threshold.

A smart contract on Ethereum may set Kleros [28] as a decentralized arbitrator system that resolves disputes. Token holders register as jurors by locking their tokens in a smart contract. The locked tokens serve as a stake. Jurors are selected randomly, while their chance to be chosen scales with the amount of tokens they lock. Their weight in the vote is proportional to the number of times they are drawn as a juror for the dispute in question and by extension it is proportional to their amount of locked tokens. The voting procedure is performed via a standard commitment scheme to prevent juror bias. Jurors whose decision agrees with the majority are rewarded with a part of the lost tokens in the stake pool as arbitration fees, relative to their weight.

We would like to refer to the reader to to [29] which analyzes advantages and limitations of token-weighted polls and using the "wisdom of the crowd" and "examines the feasibility and effectiveness of crowdsourcing information and effort using a token-weighted mechanism".

## Token-curated registries

To compensate for the transaction fees associated with storing data on the blockchain, a user has to be incentivized to publish their content. Furthermore, the need for a certain content evaluation system becomes apparent considering that it can be used not only for determining the publisher's reward, but also to determine the actual importance or value of the content. Next, we briefly present decentralized content sharing platforms and the methods they employ to ensure that the conditions mentioned above are met.

Voting is a fundamental part of the Steemit [30] is a decentralized news and blogging platform, analogous to the popular Reddit (https://www.reddit.com) platform. It contributes to publisher's revenue by rewarding posts based on its amount of *upvotes*, while voters are encouraged to curate content by gaining a percentage of the total amount earned by the post. The bigger contributors are the ones to receive the largest profit.

Various techniques are employed to mitigate Sybil attacks. Among them is the use of a special resource named *Steem Power* which can be acquired by committing an amount of the network's native, tradable coin (STEEM) to a vesting schedule for a period of 13 weeks. This resource is used to calculate a user's *Voting Power*, which essentially represents a vote's influence. Any spliting of the vesting tokens into multiple accounts will also divide their influence and thus leave the net influence intact.

A decentralized content sharing platform built on top of the Ethereum is proposed in Blockchain [31]. Users may interact with the platform by using an inflationary token named *Primas*. For instance, publishers are initially required to register by permanently locking a number of Primas tokens, while publishing content requires a small amount of tokens to be locked for seven days. In addition, a publisher may monetize her content by requiring token payments for its reproduction. Plagiarism is punished by token deductions. Users with similar interests may participate in self-governing groups whose members are responsible for evaluating the content published in the group and are subsequently rewarded for their contributions in tokens. Finally, the publishers' revenue is based on the content quality of the submitted post which is in turn measured by factors such as number of likes, reviews, recommendations and reproductions.

[32] presents a decentralized paper publication platform which allows anyone to upload and share their work in a network of researchers. Each publication creates a transaction which stores metadata such as title, keywords, publisher's address, publication transaction hashes of all cited papers and timestamp of the submission. The paper itself is uploaded to IPFS. Authors are charged with a certain amount of currency in order to submit a paper. This amount is allocated to the paper's reviewers and authors of all cited papers. After a fixed interval of time, the authors gain a reward determined by the score of the paper's reviews. From then on, the paper may only generate revenue every time it is cited. Reviewers of a paper submit a score and optionally a comment which is also stored on IPFS. Comments can be rated by readers the same way papers are rated by reviewers. Highly rated comments are rewarded.

While the above approaches seem reasonable, they neglect to mention how readers are incentivized to rate comments, assuming that rating does not come without transaction fees.

Lunyr [33] attempted to develop a crowdsourcing based decentralized encyclopedia, similarly to Wikipedia. Contribution to the network may be fulfilled either by submitting or reviewing content. Users that wish to submit are required to first review other submitted articles. They are also obligated to provide sources that prove the credibility of the information they are attempting to publish. Before an article can be successfully inserted in the platform, it must be peer-reviewed and approved by multiple users. It implements three different tokens in order to incentivize participation. LUN, a standard ERC20 token, is the reward for contributions and can be used to purchase advertising. Contribution tokens (CBN) are gained through successfully submitting or peer reviewing content and are used to determine the amount of LUN tokens that will be given as a reward to the user at the end of each two-week reward cycle. Finally, Honor (HNR) is used for governance purposes such as creating or voting in polls that concern malicious content which the reviewing system might have mistakenly approved.

We conclude this section by mentioning Thoughtcoin (https://medium.com/social-evolution/proof-of-worth-7bff8b5d85dd), Everipedia [34] and CBNT (https://web.cbnt.io) that propose solutions that are of interest but we have decided not to consider.

## Decentralized data markets

An alternative way of evaluating submitted documents is to monetize them, thereby forming a case law marketplace where document submitters have ownership of their documents and can sell them to any interested parties. A considerable amount of inspiration for this concept can be drawn from decentralized data markets. Areas of interest include the methods used for secure storage, monetization and exchange data between two or more parties, proof that the data provided by the seller is valid and the design of a transparent and efficient purchase procedure for the end-user. All these methods should ideally not compromise decentralization.

Datapace [35] is a decentralized application built on a private blockchain and based on the Hyperledger Fabric framework. It aims to create a marketplace for IoT sensor data. Buyers, sellers and validators may participate in the platform. Validators comprise a closed consortium which validates transactions using the PBFT consensus algorithm. Buyers may browse the marketplace for IoT data streams offered by sellers and receive a temporary URL to the data upon purchase. Sellers offer their collected data for sale. Datapace provides specialized hardware for sale in order to ensure that the data source is valid. Sellers who purchase the hardware are flagged as trusted, which increases their visibility in the market. Honest behavior is also incentivized due to their initial financial investment in the network.

The platform proposed in [36] allows users to monetize data collected by telemetry in a secure and decentralized manner. Users are able to set fees for the data they own and submit to the network. The data is anonymized, encrypted and finally stored in BigchainDB [37]. The data release is enacted off-chain with a key exchange between the data provider and the data consumer. Buyers are responsible for validating the data after the purchase and a reputation system is planned to facilitate the procedure.

Selected components of the proposed platform are similar to those found in a recently proposed eBook marketplace [38] which leverages blockchain technologies in a very effective manner.

Two blockchains are used for the operation of the system, `B_Chain` in which eBook information is stored and `C_Chain` which is used for the validation of purchase transactions. Authors may self-publish their work by registering it on `B_Chain`. Once validated, the eBook is split into chunks, encrypted and stored in a decentralized book repository. Direct communication and coordination between buyer and seller is required in order to successfully perform a purchase transaction. A challenge-response protocol is used in order for a buyer to determine the validity of the book's content before purchase. A random piece of the eBook is sent to the buyer for this purpose. The buyer may pay using the network's native cryptocurrency. To prevent piracy, strict Digital Rights Management (DRM) techniques are established with the use of a Service Application (SA) which re-encrypts the purchased eBook locally on the reader's machine. The reader may only access the purchased eBook through the SA. It is expected that certain design decisions taken in the development the above system may be utilized into *AnyCase* at later stages.

## Design

### System overview

We design *AnyCase*, a token-curated registry for case law. The fundamental component of the platform is a set of Ethereum smart contracts that implement the core protocol for document submission, validation, and taxonomy, as well as a tokenized economy. The platform also consists of a web application that interacts with the Ethereum blockchain and allows participation through the user's web browser.

Documents may be accessed by data consumers, such as legal professionals or casual researchers, without cost. However, participating in the platform either as a submitter or as a curator requires the utilization and management of *LAW*, the token which functions as the platform's currency and consequently, as an incentive mechanism for positive contribution to the network.

Through the platform, a user is able to upload a document to a public, decentralized and content-addressed file service such as IPFS and then submit its hash on the chain along with a deposit in LAW. Document submission triggers the creation of polls which determine its validity as a court decision, its category of law and juristic importance.

Using a commitment scheme to prevent bias, curators are able to stake LAW and submit their votes on the chain. The curators whose vote was coherent with the majority are rewarded by sharing the staked tokens of the unsuccessful voters. If a document is deemed valid by the majority, the submitter also receives part of the stake pool.

The platform incentivizes users to act as curators without relying on a centralized authority for tallying the votes or directing the voting procedure, essentially leading to the establishment of a public, decentralized and autonomous platform dedicated to case law. Furthermore, it allows both humans and machines to participate in the network. Documents and their metadata are stored in a format that is easy to parse and consume, while access to the results of the polls is publicly available via the contract's storage. Thus, the platform provides labeled data which can be used to train machine learning models. This facilitates and encourages participation by those who wish to use their machine learning model in order to earn tokens. Please note that both humans and machines may participate. Humans will find it easier to do so through the web application, while machines directly call the contract's methods after for example, running a machine learning algorithm on a document that will classify it.

## Token engineering

The platform's medium of currency is named LAW. It implements the ERC20 specification standard which is supported by the majority of exchanges and wallets, while it provides support for the management of token vesting schedules. Vesting schedules are used to turn LAW into *VLAW*, a vested token to be presented and analyzed in detail in the next section. Users may gain LAWs by

- purchasing them from the crowdsale contract,

- in exchange for Ether,

- by engaging in trading with other token holders or

- as a reward for contributing positively to the platform after successfully participating in a poll.

In addition, actions that involve voting or inserting content in the platform require an amount of LAW to be deposited in the contract as a stake. Please note that a large number of applications use tokens to measure voting rights.

**VLAW.** A vesting schedule is a method of gaining ownership of assets, such as company shares or benefits, usually after an extended period of time. Its main purpose is to encourage loyalty. For instance, an employee may earn rights to retirement benefits based on how long they have been employed in a company. A *graded vesting* schedule allows gradually gaining assets within the schedule's duration, while in *cliff vesting*, the assets are released after the duration of the cliff has passed. During the cliff, the stakeholder has no ownership of the assets.

In *AnyCase*, vesting schedules allow a user to turn LAW into *Vested LAW* (VLAW) by locking it in a contract for a predefined long period of time. The amount of VLAW gained from a vesting schedule increases linearly with time until the end of the vesting period when the full amount of VLAW has been acquired 1. Furthermore, a cliff is put into place to prevent gaining any vested tokens for a small period of time immediately after the start of the schedule. A user may revoke a vesting schedule at any time, but their LAW will be locked for a short period of time before being released.

In our platform, VLAW is an equity token which represents voting power and its main purpose is to add basic sybil resistance to the platform. Since voting power is equal to a user's

vested balance, there is no difference, for instance, between a user having 10 VLAW and 10 users having 1 VLAW each, rendering sybil attacks impossible.

Transferring VLAW means transferring voting power. Instant transferring voting power between accounts could easily lead to double-voting. The vesting cliff prevents this, due to the cliff duration being longer than the duration of the poll's commit phase. This disallows voting more than once in the same poll using the same tokens, since, by the time the vested tokens have been transferred to another account, the voting process of the poll will have come to an end.

Furthermore, the release duration should be long enough to prevent a user from intentionally vandalizing the platform and instantly selling her tokens. This ensures that the user has to wait until their actions actually affect the value of the platform and by extension, the token, before being able to sell, encouraging positive behavior. Finally, since VLAW may only be gained through long-term financial investment, it is a good indication of faith and loyalty to the platform, making it a suitable criterion for determining influence.

## Platform rules

The rules that define our platform need to be stated in detail, stored publicly and be easily accessible by all participants. As mentioned in Decentralized applications, we should avoid storing rules on the contract storage, as there is a steep cost required to store large amounts of data on-chain. Instead, the rulebook is converted to an easily parsable format such as JSON and uploaded to a public, decentralized and content-addressed file service. The service provides the hash of the file in return, which is then permanently stored on-chain during the deployment of the contract.

While there are multiple such services, which have been discussed in Decentralized applications, we have selected the IPFS service for our implementation. Therefore, for the rest of this paper, only IPFS will be mentioned. Such a service allows storing data without relying on a centralized storage provider. The rulebook is then able to be accessed and used as a reference for appropriate behavior both by submitters and curators. Furthermore, content-addressability ensures that the rulebook has not been tampered with after its submission, as that would result in its hash being different than the one permanently stored on-chain.

For the needs of our platform, we have identified the following three different types of polls:

**Validity poll**: determines whether the document in question is a valid case law or not and it begins immediately following the document's submission. The validity conditions are stated in the rulebook and should be taken into consideration by the document submitter in order to ensure that her document is accepted. If any rules are broken, an honest validator should vote against this document.

**Classification poll**: places the court decision into a category of law. This poll begins accepting votes after the end of the validity poll, on the condition that the document was deemed valid. The available categories vary depending on the predefined desired taxonomy. Examples include high-level categories such as "Labour law" and "Property law", or categories specific to the decision content such as "homicide" and "arson". The number of available categories commonly ranges from 20 to 100. The winning category is the one that gathers the majority of votes.

**Importance poll**: assesses the document's juristic value. It follows a case law ranking scheme, like the one adopted by the European Court of Human Rights [39], in which cases are placed in four categories according to their importance level. The highest ranked cases represent influential case law, while the lowest ranked cases are common and of little legal

interest. Similarly to the classification poll, this poll is scheduled to start after the end of the document's validity poll.

Other polls my be added if needed, for example, a duplication detection rule.

## Document submission

Any user may prepare and submit a court decision to the proposed platform, through its web interface and in accordance with the validity conditions stated publicly in the rulebook. Then the document and its metadata (such as the date and decision number) are converted to JSON format and uploaded to IPFS. Next, the document submission to the contract is performed following a *deposit-refund* paradigm. Specifically, the submitter sends a transaction to the contract which operates as follows:

1. Stores the IPFS hash of the uploaded document and the submitter's address in the contract in a new document entry.

2. Submits an amount of LAW as a deposit. The deposit amount is defined as a constant in the contract storage.

3. Initializes the three polls mentioned on page 13.

4. Openly votes TRUE in the newly created validity poll using the submitter's current voting power.

The submitter is surely interested in the validity poll since her deposit and reward depend on its verdict. If the document is deemed valid, the deposit is returned and the submitter may collect her part from the reward pool.

We note that it is impractical to prevent the document submitter from voting for herself. Attempting to prohibit the submitter's address from voting in the poll may result in all submitters maintaining a second account just to vote for their own documents. Furthermore, note that the submitter's vote cannot be ambiguous, as there is no incentive for someone to vote against her own document. This is also the reason why the vote is submitted openly, instead of through the commitment scheme. This adds bias to the poll, especially if the submitter is a high-influence user since this could cause other voters to vote in agreement with the submitter in order to increase their chance of gaining the reward.

## Voting process

Voting is clearly the most fundamental component of the platform. Key elements of the voting process are its commitment scheme and deposit-refund paradigm. In this section, we present and analyze the voting process and its transactional flow in detail.

A standard commitment scheme is used during the voting procedure in order to ensure the confidentiality of the vote. Confidentiality prevents bias between voters and constitutes a necessary property since voters are rewarded based on their vote and the outcome of the poll. Since transaction data in the Ethereum network is public, the vote needs to be concealed locally before sending the transaction. This prevents voters from monitoring contract transactions and copying the voting patterns of the platform's most powerful voters in order to increase their chance of gaining a reward in an ethically unacceptable way.

The deadlines of the commit and reveal phases of a poll are set upon its creation. Thus, the contract is able to compare the current block timestamp and the phase deadlines to determine the current phase. No authority is required to manually advance the poll through its phases.

The duration of each phase ought to be long enough to allow the participation of a large network population.

During the **commit phase**, voters send a transaction which:

1. Submits `hash(v, s)` to the contract, where `v` represents the vote as an index that points to the desired candidate of the poll and `s` is a locally and randomly generated value which is used as a salt. The salt is necessary since it adds entropy to the committed hash and prevents other voters from using a rainbow table (https://en.wikipedia.org/wiki/Rainbow_table) to expose the vote.

2. Deposits, as a security measure against bad actors, an amount of LAW as a stake. Those who vote incorrectly are punished by losing their stake.

3. It records the voter's weight at the time of the vote by computing their vested balance (VLAW) based on the current block timestamp.

The deposit amount is small and the same for every user in order to curb the maximum amount of tokens that one can gain from a poll. It prevents wealthy and high-influence users from gaining too many tokens by staking considerable amounts of LAW.

During the **reveal phase**, voters send a transaction which:

1. Submits the vote `v` and salt `s` to the contract and verifies that the revealed values match the committed hash by performing the hash computation on-chain.

2. Refunds part of the voter's stake in order to discourage voters from abstaining from the revealing process.

3. Adds the voter's previously recorded weight to the tally of the voted candidate, thus officially recording the vote.

As the end of the reveal phase approaches, it becomes easier to predict the outcome, with the last voters essentially having knowledge of the outcome before this is revealed to all voters. Consequently, this would lead voters to wait until the last moment to reveal their vote, or voters who predict that their vote is unsuccessful may choose to abstain from revealing in order to avoid paying the transaction's gas cost. Simply put, it would result in voter bias and a skewed poll verdict. Providing compensation for revealing leads to voters being honest and revealing their vote regardless of the final verdict.

After the end of the reveal phase, the vote tally becomes immutable as the poll stops accepting reveal transactions. The winning candidate of the poll can be determined by finding the candidate which has accumulated the most weight during the reveal phase.

Users who voted for the winning candidate may send a transaction in order to calculate and collect their stake and reward. The total reward pool consists of the staked LAW tokens of users who voted incorrectly or did not reveal their vote.

The individual reward is proportionate to the amount of the voter's VLAW that was recorded on commit. This is because sharing the reward equally between winners would hurt the protocol's sybil resistance. A user could cheat by voting with multiple accounts and earning an equal amount of tokens in all of them. In total, their share of the reward would be larger than that of honest voters who only use one account. Consequently,

$$user's\ individual\ reward = user's\ recorded\ voting\ power \times \frac{total\ reward\ pool}{total\ voting\ power}.$$

This not only prevents sybil attacks by taking into account the user's contribution to the outcome, but it also rewards high-influence voters with a larger share of the reward pool, which is a desirable property of the system.

There is no deadline for collecting poll rewards.

## Analysis

**Collective wisdom.** The ground-truth regarding the validity, category, and importance of a submitted document is unknown and the platform should not rely on a centralized authority to provide it. We ought to offer coherence in our voting results. One way to achieve it is to produce approximate ground-truth by aggregating individual knowledge.

Along these lines, [40] presents consistency with other users as a parameter that effectively evaluates human taggers. Additionally, [41] applies this concept to a blockchain which implements a decentralized data feed. It incentivizes participants to perform the same task but rewards only those whose results agreed with the majority of voters. It relies on the assumption that it is harder to coordinate and agree on a specific lie than simply being honest. [42], which implements a decentralized prediction market, also relies on this theory.

This concept may be clarified further by the following variation of the **prisoner's dilemma** mentioned in [41]. Two prisoners that are not allowed to communicate, are presented with a list of numbers and will be released if they both choose the same number.

24946 51714 100000 5453 30388

While in reality 100000 has no particular meaning in this context, it will be the most reasonable choice due to each prisoner expecting the other to also pick this number. Similarly, in our proposed platform, the rulebook constitutes the unique point of reference which encourages honesty and allows the honest majority to coordinate and receive rewards.

In the absence of the rulebook, the voting game could still be played. Nevertheless, such omission will lead to rather chaotic behavior of the voting procedure. The rulebook makes it easier for voters to coordinate on what to agree on, in order to get rewarded. If voters assume that the majority of other voters are honest, they themselves will also vote honestly for fear of losing their deposit and reward.

Individual rewards are proportional to the amount of disagreement, since, the more voters disagree, the higher the reward pool, due to more voters losing their stake. Thus, one could attempt to collude in order to increase their profits by convincing a small number of voters to vote dishonestly. For instance, the colluder announces to a small number of voters that they will vote for B while planning to vote for A. The plan is to convince a sufficient number of voters in a way that will lead to an increased reward pool but not to a different winning candidate.

Please note that an advantage of using this method to reward voters is that the rewards are provided by participants who lost. New tokens do not need to be minted to reward winners, instead, they are exchanged between the poll participants. Minting new tokens, which, concerning real-world economy, could be considered similar to printing dollar bills, would lead to inflation which could significantly affect the value of the token.

**Effects of token value.** Platform incentives are sensitive to the value of LAW. Since rewards are scaling with VLAW, the income of a low-influence voter will be significantly lower compared to other voters. When the reward of winning the poll is so low that the total sum of gas costs spent on the participation exceeds the value of the reward converted to Ether, there is no incentive to participate. This may happen when the value of LAW drops excessively.

Conversely, when the value of the token increases significantly, voters are reluctant to participate in fear of losing wealth.

These issues derive from the fact that the voting stake is set to a constant amount of LAW, which enables us to limit the reward pool. An approach to introduce reward flexibility would be to implement self-governance mechanisms in the form of general polls that affect the various parameters of the protocol, such as vote stake.

**Effects of token accumulation.** A voter or a colluding group of voters who owns more than 50% of the platform's VLAW has the ability to determine the outcome of a poll. This could easily be the case with token exchanges, which could eventually accumulate a large amount of LAW tokens and vest them in order to gain a significant amount of voting power. This constitutes a main limitation of the proposed concept.

It is impossible to limit the voting power threshold per voter in a poll since they could simply split their VLAW to multiple accounts in order to bypass it. Instead, the long vesting schedule required to gain one's full value of LAW as well as the period where LAW tokens are locked before being released from a schedule, lead to a greatly reduced token liquidity. This prevents a high-influence user from voting and instantly releasing their LAW with the intention to sell.

Assuming that their intention is to attack the platform, their negative contribution would cause the platform to be considered unpredictable and unreliable, therefore decreasing the value of the token; honest participants will leave or choose not to join the platform in the future. Since the attacker is not able to instantly sell her tokens, her attack would harm her wealth. Consequently, the protocol assumes that the curators with the highest amount of influence in the network are honest and attempt to contribute positively to the network out of self-interest.

In order to mitigate the initial influence of users, a *user score* parameter is introduced, which is initialized to zero and increases with each positive contribution. The way a contribution is counted scales with a user's VLAW in order to avoid sybil attacks. Otherwise, a person that votes with two accounts earns two contributions instead of one.

The user score could represent a multiplier that is applied to a voter's VLAW with every vote, adjusting their power. In addition, the user score has to increase indefinitely. If a cap on the user score was applied, it could be easily bypassed by creating another account every time the cap is reached on the previous account.

However, the multiplier will eventually reach the threshold where it no longer reduces voting power. This would lead to the user gaining even more voting power, which will, in turn, lead to an even higher user score. A VLAW-wealthy user could very quickly increase her user score due to the user score gains scaling with VLAW. Thus, the user score parameter would significantly decelerate the progress of low-influence users, while a high-influence user could bypass the handicap with a limited contribution.

**Reliability.** While crowd-sourcing image labeling or sentiment analysis could yield satisfactory results regardless of the voters' technical skills or specialties, the validation and classification of case law demand extensive knowledge of legal terms. Thus, uncertified voters may not be considered adequate for the task since they cannot guarantee to have the knowledge to curate the submitted content.

Sacrificing decentralization is unavoidable, in present times, in order to gain authenticity and reliability. A semi-decentralized solution could rely on decentralized identifiers (DIDs) and blockchain certifications for legal knowledge provided by institutions in order to authenticate voters. Authenticated legal professionals could be provided with a user profile that starts with a high influence. There has already been a related initiative to store academic records on the blockchain [43–45].

**Exploring the marketplace concept.**   Authoring and submitting a document requires a user to expend a lot of effort and the reward gained from the voting process might not be enough to justify that effort. The formation of a decentralized market for case law is a possible approach that could infuse the submitted documents with financial value.

Such approaches are presented in [35, 36, 46] and come with consequences to convenience, financial cost, and decentralization. More importantly, the marketplace concept faces a more serious problem when it comes to preserving the security of the documents since access to them by curators is necessary for their validation before being made available for purchase.

A method to mitigate the problem would be to require users to purchase the document before they are able to process it. This, however, would lead to users being reluctant about which documents they choose to process, preferring documents in which they have a direct interest, thus reducing the overall participation in the network. As a consequence, the amount of attention a document receives would be directly proportionate to the interest the network shows for it. This would lead to some documents not being processed correctly due to low participation.

Furthermore, piracy and collusion between buyers can also result in a loss of income for the submitter. Buyers may freely resell or share a document after its purchase. Additionally, they could collude by agreeing to share the cost of the document.

The data exchange between the two parties, *buyer* and *seller*, should be performed using a secret channel. However, since off-chain communication and data flow is not verifiable, disputes could arise. The seller could choose to deliberately send invalid data. In that case, the buyer would have to initialize a dispute and the seller would have to prove their innocence. Furthermore, the buyer could receive the correct data but attempt to grieve the seller by falsely accusing them. Dispute resolution would then require the document to be revealed to the arbitrators in order to reach a decision.

The aforementioned issues add significant complexity and inefficiency to the marketplace. In conclusion, the lack of verifiability of intra-user transactions, uncertainty concerning the real content of a document due to it being hidden before purchase, reliance on a reputation system to ease this uncertainty, constitute limitations which result in a complex and inefficient marketplace.

We summarize the description of our design efforts in Fig 1 where the components and the actors involved in our *AnyCase* ecosystem are depicted and are interrelated.

## Implementation

In this section, we present a proof-of-concept implementation of the protocol described in section Design.

### Smart contracts

Three smart contracts have been written in Solidity, the Ethereum's smart contract programming language; the main contact, the token contract, and the crowdsale contract.

The Truffle framework [47] has been used during the development to generate the project template, as well as to provide tools for project management such as developing, unit testing and deploying the contracts to both private and public test networks. During early development, Ganache [48], a personal Ethereum blockchain simulator, was used to launch a local, private blockchain for testing contract interactions and monitoring transactions.

**Main contract.**   The main contract stores most of the vital application data and implements the document submission and voting protocol. The main data structures used throughout the contract are `Document` and `Poll`.

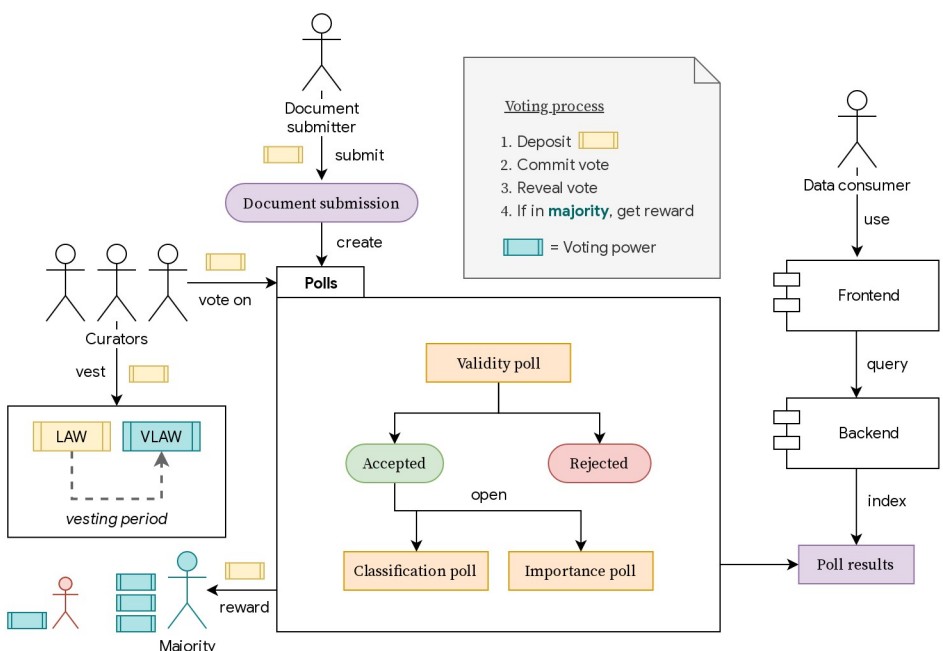

**Fig 1. Components and actors.** Functional components and actors in the *AnyCase* ecosystem.

The `Document` struct stores only the submitter's address and the IPFS hashes of the uploaded document. Documents are stored in a global `documents` array, as the web application might need to access them sequentially, in order, for instance, to display the 10 most recent documents.

The `Poll` struct stores its poll type, the index of the document it is associated with, deadlines for the commit and reveal phases, and a helper `VoteData` struct which includes vote information such as committed hash, vote weight and revealed vote for every user. Each piece of information is stored in a `mapping`, a key-value table that maps addresses to information. We use the helper struct merely due to a limitation in the Solidity language, which does not allow returning arrays of structs that contain `mapping` fields. Wrapping all the mapping fields in a separate struct bypasses this limitation.

It is worth pointing out that, the sole functional difference between the poll types, as far as the contract concerns, is their number of candidates. Thus, the contract requires only the number of available candidates of each poll type to be recorded in its storage. This allows the contract to initialize the vote storage of each poll, represent each candidate as an index which corresponds to the desired candidate in the candidate array and finally, to verify whether a submitted vote is valid or not during the reveal phase. The contract is agnostic of the meaning and verbose description of each candidate. These may be stored in the rulebook, which can be easily fetched from IPFS and displayed in the web application.

Similarly to `documents`, a global `polls` array is initialized. New documents and polls created by `addDocument()` are appended to their respective arrays. Thus, knowledge of the poll index and voter address allows $O(1)$ access to a voter's data for the poll in question.

The contract data could be linked in more complex ways to allow the contract to answer queries such as "documents which were uploaded by user $0x1234$ between April and June". However, that would increase storage needs and by consequently, gas costs for the user. In general, it is adequate to use arrays of `struct`s for data that needs to be accessed sequentially

and `mappings` for data that needs to be accessed by an arbitrary key. To answer more complex queries, an application which indexes and aggregates contract events is required, such as TheGraph [49].

The commitment scheme is implemented as described in Voting process. The functions `commitVote()` and `revealVote()` are used during the commitment scheme. The modifiers `canCommit` and `canReveal` are applied to the aforementioned functions in order to test if the conditions for a valid commit or reveal are satisfied. If this is not the case, the transaction is reverted. In addition, the modifier `withTokenDeposit()` is to be applied to functions `addDocument()` and `commitVote()` since they require a user to stake tokens to proceed. Finally, `collectReward()` determines whether a user is eligible for collecting a reward for a specific poll, calculates the reward based on the method described in section Voting process and transfers the tokens to the transaction sender.

**Token and crowdsale contracts.**   OpenZeppelin [50] provides contract templates, math libraries and token implementations for common token specifications. Our work was based on the implementation of OpenZeppelin's ERC20 token. The full token name is "LAW Coin" and its symbol is "LAW". It is represented with 18 decimals to mimic the relationship between Ether and wei.

The token ties itself to the platform by allowing the address of the main contract to make transfers to itself through an additional helper function, `dappTransfer()`. Due to the specification of ERC20 tokens, for a contract to transfer tokens on behalf of a user, two transactions are required: an `approve()` transaction followed by a `transfer()`. Thus, with `dappTransfer()`, the token contract can authorize the main contract to make transfers to itself on behalf of users using only one transaction.

The creation of a vesting schedule essentially determines the parameters that define the line seen in Fig 2. These parameters are stored in a `VestingSchedule` data structure and each user is mapped to an array of such structures. To calculate the VLAW balance gained so far from a vesting schedule, we need to determine the value of the line function, given the current block timestamp. Consequently, the total vested balance of a user is the sum of their vesting schedule balances.

A loop is needed to compute the sum of balances. However, loops are not ideal due to high gas costs. If the vesting schedule duration is short enough, it is possible to achieve lower gas consumption during transactions that need to compute a user's vested balance by occasionally restructuring the vesting schedules array to decrease its size. Completed vested schedules no longer need to be computed. During the restructuring, it is possible to delete them from the array and keep their balance in a separate variable. Deletion is achieved by swapping the element to be deleted with the last element and then decreasing the length of the array. This

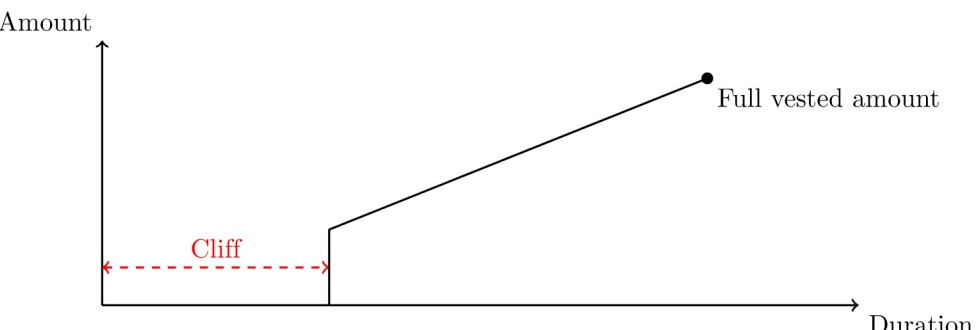

**Fig 2. Example of a token vesting schedule using a combination of cliff vesting and graded vesting.**

causes an amount of gas to be refunded to the transaction's sender, since the transaction deletes data from a contract's storage.

Finally, our crowdsale contract is also based on the crowdsale implementation offered by OpenZeppelin. A standard non-timed crowdsale with a maximum total supply provided and a stable conversion rate of Ether to tokens was used and no additional business logic has been implemented.

Multiple unit tests were implemented to ensure the correctness of the three aforementioned contracts. They were performed in a locally simulated blockchain, using Ganache, which allows complete manipulation of block timestamps. This manipulation is not possible in public Ethereum networks, making them an unsuitable testbed for simulations of long-lasting procedures like vesting.

**Time-dependent contract logic.** The commitment scheme which is essential for ensuring a privacy-preserving voting procedure requires time-dependent logic to function successfully. The same applies to the token vesting schedule functionality implemented in the token contract. In Solidity, special global variables such as `block.timestamp` and `block.number` are made available for this purpose.

Relying on the timestamp of the block is a possible solution. However, its accuracy is questionable as the miner can adjust it to some degree and is expected to do so if there is a large enough profit to be gained by this action. As a more secure solution, [15] recommends using the block number in conjunction with the average block time of the chain to estimate time, since it is more difficult for a miner to manipulate the block number.

The Ethereum network attempts to keep the block time at 15 seconds on average [51]. The block difficulty is adjusted accordingly to achieve this. However, the transition of Ethereum to a Proof-of-Stake network could decrease the block time, potentially rendering any estimations inaccurate. Furthermore, in this contract, no critical decision is made by relying on the block's timestamp. Plenty of time is provided for all time-sensitive actions of the contract. Additionally, regarding vested balance which is also time-dependent, a deviation of several seconds would be negligible when considering the months or years required to complete a vesting schedule. Due to the aforementioned points, we opt for the timestamp method to enforce time constraints.

## Web application

The web application provides users with an intuitive interface to the deployed contracts and their methods. Through the frontend of the application, it is possible to manage their tokens and vesting schedules, as well as participate in polls and submit documents. The frontend is developed as a Single-Page Application (SPA) using the ReactJS framework, meaning that all routing happens on the client-side. Due to this fact, the frontend of the application can be hosted on IPFS, achieving, as a result, a higher degree of decentralization and potentially accelerated access as its popularity increases.

The backbone of the application is `web3.js`, a JavaScript library that allows interaction with an Ethereum provider through an HTTP or IPC connection. The provider may be either a local node running with `geth` or a remote node offered by various services such as Infura [52].

The MetaMask browser extension, a non-custodial wallet and Ethereum light-node, is used for interacting with the applications by signing and sending the user's transactions to the network. The application is tightly integrated with MetaMask as it is the least intrusive wallet solution for a new user and can be used through all major web browsers.

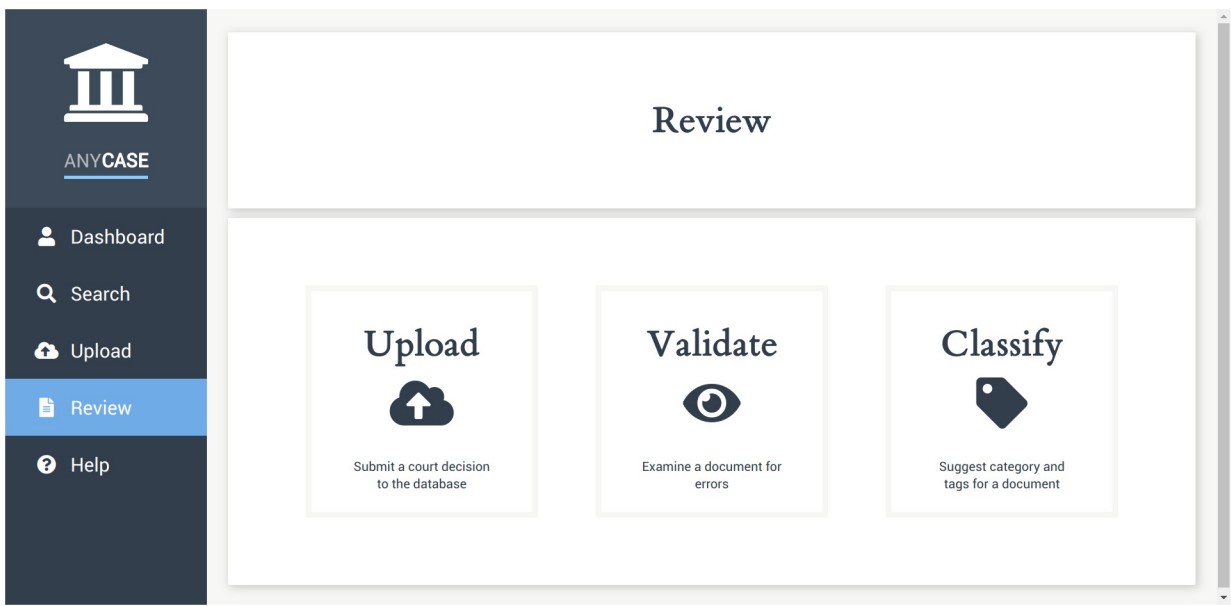

**Fig 3. Snapshot of an early version of the user interface.**

The user may navigate, as depicted in Fig 3, through the following pages:

1. The **upload** page allows a user to submit a document for validation. It provides forms for various metadata such as court decision number and headnote, which is a summary of the case separated from the document body. Documents are submitted to IPFS in JSON format, which is easy to parse and is widely used by most APIs. Finally, the `addDocument()` contract method is called in order to perform the actions mentioned in Document submission.

2. The **search** page allows a user to submit queries and browse through submitted documents. The search results display the headnote of the case as well as metadata such as category and importance level, if the respective document polls have ended. Rendering a selected document involves fetching it directly from IPFS and parsing the returned JSON.

3. The **contribute** page allows a user to participate in any of the three polls mentioned in Platform rules. Upon choosing the desired poll type, all the open polls of such type are fetched and rendered as a list. Clicking on a poll fetches its data from the contract, including the document it is associated with. If the user can vote for the poll, the document is fetched directly from IPFS and displayed along with the poll's question. For instance, in the case of a validity poll, the validity conditions are displayed alongside the document and the user is instructed to tick them off individually. A cryptographically secure, randomly generated salt is created with `web3.utils.randomHex()` and finally, `commitVote()` is called. The vote and salt are stored in the user's `localStorage` on transaction confirmation to be revealed later on.

4. The **dashboard** page allows a user to manage their LAW and VLAW as well as post-commit voting procedures. It fetches the pending polls in which a user has committed their vote and, depending on their state, displays information about their reveal deadline and whether or not the user is eligible to collect their reward. Furthermore, a mechanism for exporting and importing the committed data is available. Its main use case is to back up locally stored vote data and import it to different browsers or computers.

5. The **help** page provides information about the platform in a Frequently Asked Questions (FAQ) format.

## Storage and IPFS limitations

The main disadvantage of content-addressed file services is their lack of support for sequential storage. Thus, files need to be downloaded individually to be processed in a meaningful way. Aggregation of data is not possible natively, thus, this functionality has to be implemented as a separate application on top of the file service.

To accommodate the needs of data consumers, it is necessary to maintain a server, or a server cluster, that aggregates and indexes documents and accepts full-text search queries from users. This will constitute the only centralized component of the platform. It is noted that this functionality is only necessary to cover the needs of data consumers, who do not participate in the curation process, but are only interested in browsing through documents.

Additionally, IPFS cannot guarantee data availability, as non-popular data is in danger of being garbage collected from the network. One or multiple IPFS nodes that pin newly added documents are required to ensure high availability and access speed for all documents.

## Legal limitations

Legal issues are commonly raised during the design, implementation and operation of any decentralized system that utilizes Blockchain as a Content Distribution Technology. Legal issues like copyright and display right have been considered early in the blockchain era. Several of them still challenge researchers and developments. The system proposed in this paper has certain characteristics that allow us to handle, to a certain extent, such challenges. Specifically, we note that our system couples Blockchain with IPFS for content distribution and as such it provides anonymity (pseudo naming) and each node only stores data that has been explicitly and knowingly shared by the submitter for the purpose of being publicly reviewed in exchange for a potential reward. This leads to a relatively stable legal state by essentially eliminating most of the legal, and censorship issues. For the rest of the issues, for example the right to forget, we ought to follow recent and on-going developments [53, 54], which are in fact orthogonal to our study.

Anonymization of case law documents is another significant matter that needs to be addressed. While it has been noted that anonymization can make legal research harder [7, 55], countries which have adopted strict data protection laws are making efforts to remove identifying information such as names of physical persons occurring in the bodies of court decisions. Several countries aim to completely anonymize case law and have already kick-started the process of upgrading their anonymization tools for this purpose [56, 57].

Consequently, depending on the platform's country of operation and the data protection and privacy regulations it is required to adhere to, document submitters might be forced to anonymize the data they provide. In such cases, anonymization should be added to the platform's document validity conditions. Furthermore, a method of de-personalization should be employed to preserve both the privacy of the involved parties and the usefulness of the data.

In order to protect the privacy of the involved parties, de-identification could be achieved by replacing victim and defendant names with initials such as V1, V2 and D1, D2 respectively. Genders of physical persons would have to be preserved as they may constitute a crucial parameter of the verdict, such as in cases concerning divorce and custody.

Our overall proof-of-concept implementation efforts are summarized in Fig 4 which illustrates the full stack of our proposed decentralized system, the technologies considered in the

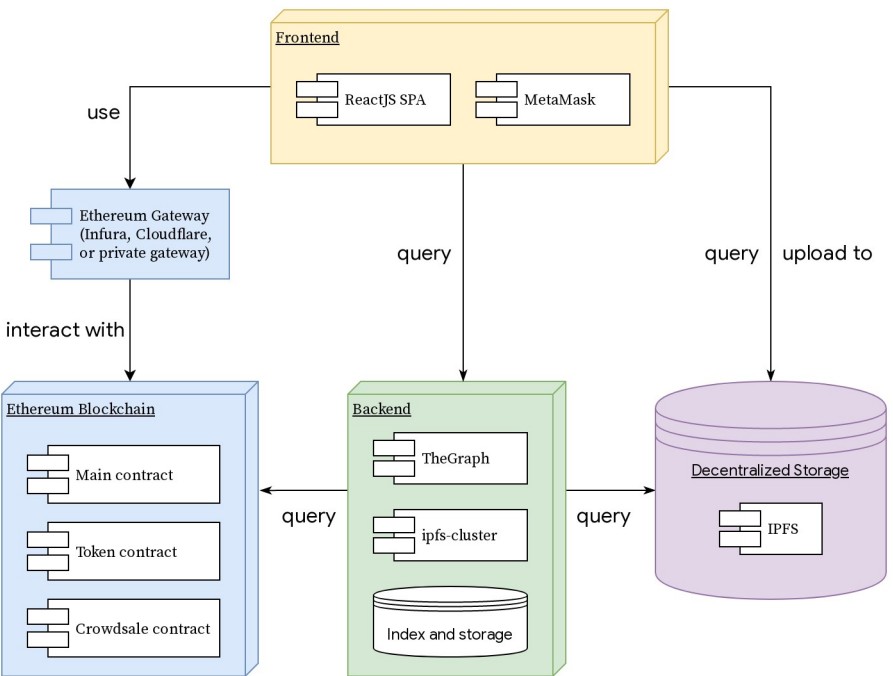

**Fig 4. Technologies and tools.** Enabling technologies and tools associated with the *AnyCase* ecosystem implementation.

previous three sections and used to achieve its implementation and the manner in which these technologies interact.

## Efficiency

Gas is the most substantial indicator of smart contract efficiency in the Ethereum network. Low gas costs denote efficient contract storage usage and minimal computation needs, which directly translate to lower transaction fees for the end user.

The focal point of our platform is the decentralized voting game used to achieve consensus. In this game, transaction fees constitute an inevitable cost and a necessary risk that its players have to take in order to participate. An analysis of gas costs will facilitate the detection of risk and reward disparities between the platform's users and help adjust various parameters of the protocol, such as the stake amount and the reward calculation formula.

A gas cost analysis has been performed on all the available actions of the platform from the perspective of a single end user. In Fig 5, each column represents an action of the platform, while each action is broken down into all the contract methods required to complete the action in question. It is noted that each contract method corresponds to a single transaction sent by the user to the contract either manually or through the web interface. The base transaction cost of 21,000 gas is included in all measurements.

In Table 1, the total gas cost for every action is also converted to Ether and USD. It is noteworthy to mention that, due to the volatility of Ethereum, the Ether to USD exchange rate and the average gas price required for a medium-priority transaction could change dramatically within a short span of time. For instance, network congestion may force users to pay a higher gas price to avoid facing slow transaction confirmation times.

With regard to the token vesting action, computing its total cost is inherently problematic since it is largely dependent on user behavior. The data attempts to present a cost analysis of a

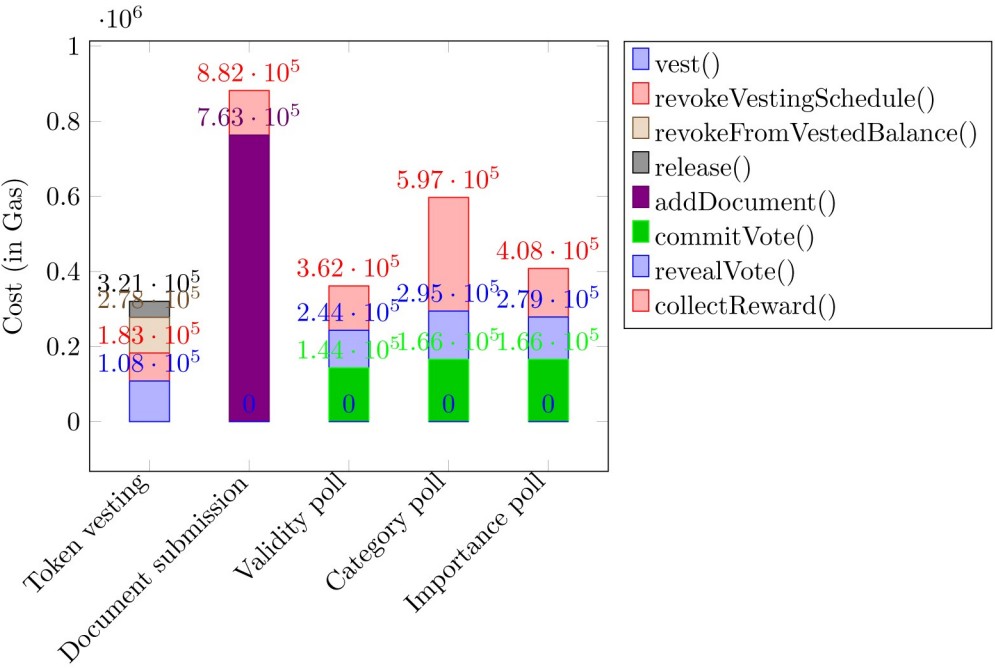

**Fig 5. Contract methods and their respective gas costs, grouped by platform action.**

simulated token vesting round where the user vests tokens, then revokes a vesting schedule and an amount of fully vested VLAW, and ultimately releases their tokens. In reality, certain users may never wish to release their tokens, preferring to continuously gather influence. This behavior is highly encouraged by the protocol, mainly due to the enforced vesting cliff and token release delay, as mentioned in section VLAW. Other, more opportunist users, may attempt to maximize their profits by revoking their schedules if they predict that the token's value will be high by the time their tokens are released. Consequently, in the case of token vesting, it would be more sensible to observe the gas costs of individual contract methods, rather than their sum.

The document submission data demonstrates that submitters are severely affected by high gas costs. This is anticipated, as the burden falls on the shoulders of the submitter to initialize data structures, create polls and cast her positive vote in the newly created validity poll, all in one transaction. This causes a large amount of data to be written in the contract storage which leads to an increased gas cost compared to other contract methods. To ensure that submitting documents is worthwhile, the reward calculation algorithm could be adjusted in order to

**Table 1. The combined gas costs of all transactions required to complete each platform action, from the perspective of a single user.** The exchange rate of Ether to USD has been set to $155 and the gas price to 15Gwei (real values recorded on April 6th 2020). [58, 59].

| Action | Cost | | |
|---|---|---|---|
| | in Gas | in Ether | in $ |
| Token Vesting | 320716 | 0.0047 | 0.73 |
| Document Submission | 881547 | 0.0133 | 2.06 |
| Validity Poll | 361651 | 0.0055 | 0.85 |
| Category Poll | 596901 | 0.0089 | 1.38 |
| Importance Poll | 408035 | 0.0061 | 0.95 |

account for this difference in gas costs and allocate a larger share of the reward to the successful document submitter as compensation for the higher risk. This would also reflect the higher amount of time and effort required to submit rather than review a document. In cases when the document is deemed invalid, the submitter's additional reward should remain in the contract and not be distributed to the winners, since this could lead to voters deliberately voting against the submitter in order to receive a larger reward.

The gas cost of `commitVote()` scales with the user's currently active vesting schedules, since it calculates and stores her vested balance in the poll as her vote weight. Each active vesting schedule is stored as an entry in the user's personal vesting schedule list. Computing the balance requires looping over the schedules, which indicates that the gas cost of computing a user's vested balance could scale poorly if she decided to vest her tokens by creating a significant number of low-value schedules. To prevent this, the `vest()` method could be adjusted to require a minimum amount of tokens to be locked in order to successfully create the vesting schedule, which would force users to preserve their tokens and create fewer, but high-value vesting schedules. In our simulation, the user created five vesting schedules and committed their vote shortly after their cliff point.

The gas cost of `collectReward()` scales with the number of candidates in the poll, since, in order to allow a user to collect their reward, it needs to compute the verdict and check if the user's vote agrees. The verdict is computed by looping over all candidates to determine the winning category. Our simulation set the number of law categories to thirty-six and the number of importance levels to four. While the presented costs are acceptable, it is worth investigating if, for a much larger number of categories, it would be wiser to make each user check and update the winner inside `revealVote()`. Particularly, it would be beneficial to calculate the exact threshold of candidates after which the aforementioned method becomes more efficient than the current one. It would allow `collectReward()` to have a constant gas cost regardless of the number of candidates in the poll, as the verdict would be computed and finalized by the end of the reveal phase. The side-effects of this optimization include an increase in gas costs both for the document submitter and the voters, since more data would need be stored in each poll entry and more costly `SSTORE` operations would need to be performed to update the winner field of the poll during vote reveal.

## Security

Smart contract security is a crucial aspect of any decentralized application. Contracts are permanent, immutable and are responsible for managing currency. Additionally, there is no way of updating a smart contract, besides destroying it by sending a `selfdestruct()` transaction. Therefore, it is of utmost importance to ensure the correctness of a contract before its deployment to a public network.

Regarding the detection of contract vulnerabilities as well as testing the contract against malicious users, the Main Ethereum network would constitute the ideal testbed since attackers are regularly attempting to exploit contract vulnerabilities for their profit. However, deploying and testing the contract would cost real currency and would not allow rapid and continuous testing. As such, testing on the Main Ethereum network constitutes a costly and impractical process. Consequently, the majority of smart contract developers instead opt for public Ethereum test networks such as Ropsten, where test Ether can be obtained without exchanging real currency.

However, due to our contracts being largely time-dependent, it is nigh infeasible use any public network as a testbed. Instead, we have relied on comprehensive unit testing to ensure that all the protocol rules are correctly enforced. For this purpose, we have selected the Mocha

test framework, which, in conjunction with the Truffle framework allows us to create unit tests which deploy our contracts and send transactions which attempt to cheat and break the protocol, while observing whether the contract is able to detect and revert these transactions. Unit testing in a local blockchain allows the manipulation of the chain's `block.timestamp`, which allows us to test the soundness of time-dependent actions such as token vesting and transitions between poll phases.

## Synopsis and prospects

This paper introduces *AnyCase*, a decentralized case law curation ecosystem powered by intrinsic financial incentives that generate wealth for its benefactors.

As seen in section Design, while commitment schemes are well-established components of private voting in trust-less environments, the majority of voting protocols rely on an administrator entity to manage the voting phases and compute the tally.

Our overall voting protocol relies solely on contract-enforced time constraints which are initialized on document submission, while the tally is computed individually by every voter, making the voting protocol and process completely decentralized and autonomous since it requires no tallying authority or administrator entity to operate.

The voting process itself is privacy-preserving and publicly verifiable, while it allows voter abstinence in all phases and does not compute the tally based on the number of individual voters, rendering it completely sybil-resistant.

Lastly, no secret channels between participants are necessary during the voting process. Communication is only performed between the user and the smart contract through transactions.

The implementation and analysis of the *AnyCase* concept are by no means complete. Due to the expertise needed to perform the tasks identified in the platform, it is necessary to raise the average skill level of participants and thus the reliability of the platform as a knowledge base. A semi-decentralized platform that identifies and authorizes skilled curators might result in higher quality validation and classification. Unfortunately, identity on the blockchain is one of the hardest problems to solve without sacrificing decentralization.

Centralized systems are well established and, to some extent, are considered as de facto standards while decentralized ones are rapidly emerging, disturbing almost all service sectors and beyond. During the past decade the dichotomy of these two approaches has been widely accepted and the associated strategies were considered as mutually exclusive. Late studies challenge this thinking claiming that both approaches may very well operate in harmony. Before we invest on hybrid collaborative systems involving centralized and decentralized case law sub-systems, research efforts are needed to investigate certain concepts, elucidate others and overcome technical obstacles [60]. Such efforts are underway, they are beyond the scope of this study and will be presented elsewhere.

Our paper places substantial emphasis on the careful design of the rules and incentives that define the protocol as well as the combination of technologies and components that establish the platform's entire solution stack. The suggested metadata structure of a legal document used in our paper serves as a flexible base on which our proposed protocol can be built. Any additional research in that area would complement but not invalidate our work.

A more in-depth financial analysis would be required to determine the solidity of the market this platform creates for the LAW token. It is vital to investigate the gas costs associated with participating in the platform in order to set sensible values for various parameters, such as the conversion rate from ether to LAW. Such investigation is underway. Furthermore, self-governance mechanisms should be researched to increase autonomy and resistance to factors

such as the token's market value and the value of Ether, which, for the time being, directly affect the incentives of the protocol.

Close collaboration with legal professionals is crucial in order to identify the key elements which assist legal research through an online case law database. It is also vital to determine the pain points of user experience in order to design a practical and intuitive user interface for audiences unfamiliar with cryptocurrency and decentralized applications. To achieve the above and to elucidate several blockchain related issues a pilot study is needed. Such a study is beyond the scope of this paper.

Finally, it would be interesting to investigate the feasibility of using such an ecosystem as an oracle for a next-generation decentralized court.

## Acknowledgments

The authors would like to thank Prof. Marios Haidarlis for proposing the basic concept behind our case law system and Mr. Christos Poulios for his useful comments and domain expertise.

## Author Contributions

**Conceptualization:** Eleni Panagou, Manolis Vavalis.

**Formal analysis:** Eleni Panagou, Manolis Vavalis.

**Methodology:** Manolis Vavalis.

**Project administration:** Manolis Vavalis.

**Software:** Eleni Panagou.

**Supervision:** Manolis Vavalis.

**Writing – original draft:** Eleni Panagou.

**Writing – review & editing:** Eleni Panagou, Manolis Vavalis.

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
