## [Decision Letter · Decision Letter 0]

22 Jul 2020

PONE-D-20-09822

Towards an Open and Decentralized Case Law Curation Ecosystem

PLOS ONE

Dear Dr. Vavalis,

Thank you for submitting your manuscript to PLOS ONE. After careful consideration, we feel that it has merit but does not fully meet PLOS ONE’s publication criteria as it currently stands. Therefore, we invite you to submit a revised version of the manuscript that addresses the points raised during the review process.

We look forward to receiving your revised manuscript.

Kind regards,

Wenbo Shi

Academic Editor

PLOS ONE

Journal Requirements:

2. Please remove your figures from within your manuscript file, leaving only the individual TIFF/EPS image files, uploaded separately.  These will be automatically included in the reviewers’ PDF.

3. Please upload a new copy of Figure 1 as the detail is not clear. Please follow the link for more information: https://blogs.plos.org/plos/2019/06/looking-good-tips-for-creating-your-plos-figures-graphics/" https://blogs.plos.org/plos/2019/06/looking-good-tips-for-creating-your-plos-figures-graphics/

4. Please ensure that you refer to Figure 4 in your text as, if accepted, production will need this reference to link the reader to the figure.

Reviewers' comments:

Reviewer's Responses to Questions

**Comments to the Author**

1. Is the manuscript technically sound, and do the data support the conclusions?

Reviewer #1: Yes

Reviewer #2: Partly

2. Has the statistical analysis been performed appropriately and rigorously? 

Reviewer #1: No

Reviewer #2: I Don't Know

3. Have the authors made all data underlying the findings in their manuscript fully available?

Reviewer #1: Yes

Reviewer #2: Yes

4. Is the manuscript presented in an intelligible fashion and written in standard English?

Reviewer #1: Yes

Reviewer #2: Yes

5. Review Comments to the Author

Reviewer #1: The study proposed a very interesting techno-legal which has potential to address the problem of centralized legal case repositories, and aid in the addressing some legal bottlenecks. However, the research content of the manuscript is largely ignored. For instance,

1. the problem of unified format ("since no standard metadata structure has been agreed to represent a case law document") for case law representation is largely ignored. This could have formed a major component of the research, upon which the development can be evaluated.

2. in the related works section, the study identified important literature in decentralized platforms, but largely ignored legal platforms, either decentralized or centralized. This would have formed the basis of comparison in the discussion section, Again, no such comparison is provide in the discussion section.

3. Evaluation of the system is largely missing in the report. Evaluation in this case can leverage software engineering approach, as well as information system approach (where usability, relevance, content reliability, legal limitation, as well as potential challenges). This is where significant result is expected to be generated to validate and justify the study.

4. The section on the reliability of the decentralization is vaguely presented. The study further attempted to highlight the need for a semi-decentralized platform. This section is ambiguously presented, and it constitutes a major research concern.

5. One would expect to see a comparative analysis of some sort, given the wide array of decentralized platform, and the legal case work-house identified in the introductory section of the manuscript.

Reviewer #2: The manuscript idea is novice and interesting. The author(s) proposed a public and

decentralized platform that allows the submission of court decisions in a decentralized

database. However, some design decisions need to be supported and some points need clarification. For example, why authors used Ethereum not waves or bitcoin. How system performance and security is assessed? How the system is compared to other systems? Are there any quantitative data we can rely on for the system assessment?

6. PLOS authors have the option to publish the peer review history of their article (what does this mean?). If published, this will include your full peer review and any attached files.

Reviewer #1: No

Reviewer #2: No

---

## [Author Response · Author response to Decision Letter 0]

6 Sep 2020

Editor:

We have followed all the requirements and corrected our paper. Thank you.

Reviewer 1:

We have made certain changes to our paper and replied to all your points.

Thank you for all your insightful comments and suggestions.

Reviewer 2:

Your questions and suggestions were helpful and on point.

We have made certain changes to our paper and replied to all your points. Thank you.

---

## [Decision Letter · Decision Letter 1]

18 Sep 2020

Towards an open and decentralized case law curation ecosystem

PONE-D-20-09822R1

Dear Dr. Vavalis,

We’re pleased to inform you that your manuscript has been judged scientifically suitable for publication and will be formally accepted for publication once it meets all outstanding technical requirements.

Kind regards,

Wenbo Shi

Academic Editor

PLOS ONE

Additional Editor Comments (optional):

Reviewers' comments:

Reviewer's Responses to Questions

**Comments to the Author**

1. If the authors have adequately addressed your comments raised in a previous round of review and you feel that this manuscript is now acceptable for publication, you may indicate that here to bypass the “Comments to the Author” section, enter your conflict of interest statement in the “Confidential to Editor” section, and submit your "Accept" recommendation.

Reviewer #1: All comments have been addressed

2. Is the manuscript technically sound, and do the data support the conclusions?

Reviewer #1: Yes

3. Has the statistical analysis been performed appropriately and rigorously? 

Reviewer #1: N/A

4. Have the authors made all data underlying the findings in their manuscript fully available?

Reviewer #1: Yes

5. Is the manuscript presented in an intelligible fashion and written in standard English?

Reviewer #1: Yes

6. Review Comments to the Author

Reviewer #1: The revised manuscript has fully addressed my initial observations. Painstaking care has been considered in expanding the discussion section which further integrate the security point of view. This in my opinion, is a good improvement.

7. PLOS authors have the option to publish the peer review history of their article (what does this mean?). If published, this will include your full peer review and any attached files.

Reviewer #1: No

---

## [Editor Report · Acceptance letter]

23 Sep 2020

PONE-D-20-09822R1 

Towards an open and decentralizedcase law curation ecosystem 

Dear Dr. Vavalis:

I'm pleased to inform you that your manuscript has been deemed suitable for publication in PLOS ONE. Congratulations! Your manuscript is now with our production department. 

Kind regards, 

on behalf of

Dr. Wenbo Shi 

Academic Editor

PLOS ONE